# Associations of hypertension and antenatal care-seeking with perinatal mortality: A nested case-control study in rural Bangladesh

**Allyson P. Bear**[1¤]\*, **Wendy L. Bennett**[2], **Joanne Katz**[1], **Kyu Han Lee**[1], **Atique Iqbal Chowdhury**[3], **Sanwarul Bari**[3], **Shams El Arifeen**[3], **Emily S. Gurley**[1]

**1** Department of International Health, Johns Hopkins Bloomberg School of Public Health, Baltimore, Maryland, United Stated of America, **2** Division of General Internal Medicine, Department of Medicine, Johns Hopkins School of Medicine, Baltimore, Maryland, United Stated of America, **3** Division of Maternal and Child Health, International Centre for Diarrheal Disease Research, Bangladesh, Dhaka, Bangladesh

¤ Current address: International Development Division, Abt Associates, Rockville, Maryland, United Stated of America

\* allyson_bear@abtassoc.com

**Data Availability Statement:** Relevant variables for this analysis are available from the Dryad database (https://doi.org/10.5061/dryad.9kd51c5mk).

## Abstract

Maternal hypertension may be an underrecognized but important risk factor for perinatal death in low resource settings. We investigated the association of maternal hypertension and perinatal mortality in rural Bangladesh. This nested, matched case-control study used data from a 2019 cross-sectional survey and demographic surveillance database in Balia-kandi, Bangladesh. We randomly matched each pregnancy ending in perinatal death with five pregnancies in which the neonate survived beyond seven days based on maternal age, education, and wealth quintile. We estimated associations of antenatal care-seeking and self-reported hypertension with perinatal mortality using conditional logistic regression and used median and interquartile ranges to assess the mediation of antenatal care by timing or frequency. Among 191 cases and 934 matched controls, hypertension prevalence was 14.1% among cases and 7.7% among controls. Compared with no diagnosis, the probability of perinatal death was significantly higher among women with a pre-gestational hypertension diagnosis (OR 2.90, 95% CI 1.29, 6.57), but not among women with diagnosis during pregnancy (OR 1.68, 95% CI 0.98, 2.98). We found no association between the number of antenatal care contacts and perinatal death (p = 0.66). Among women with pre-gestational hypertension who experienced a perinatal death, 78% had their first antenatal contact in the sixth or seventh month of gestation. Hypertension was more common among rural women who experience a perinatal death. Greater effort to prevent hypertension prior to conception and provide early maternity care to women with hypertension could improve perinatal outcomes in rural Bangladesh.

## Introduction

Hypertensive disorders of pregnancy include maternal chronic hypertension, pregnancy-induced hypertension, pre-eclampsia, and eclampsia. Globally, these conditions are estimated

**Funding:** This work was supported by the Bill & Melinda Gates Foundation, Seattle, WA (https://www.gatesfoundation.org/) under grant number OPP1126780 held by SEA and ESG. The funder of the study had no role in the study design, data collection, data analysis, data interpretation, the writing of the report, and in the decision to submit the article for publication.

**Competing interests:** The authors declare that they have no competing interests.

to complicate 5–10% of all pregnancies and are responsible for 16% of the estimated 2.6 million stillbirths and 50% of the estimated 2.9 million neonatal deaths each year [1–6]. In 2010, an estimated 224 million women of reproductive age across the world had chronic hypertension [7].

Between 2011 and 2018, the prevalence of hypertension in Bangladeshi women over the age of 35 increased from 32% to 45% [8, 9]. In 2018, the prevalence of hypertension in women aged 18–34 was 12.5% [8]. The burden of pregnancy-induced or chronic hypertension in pregnancy is less well understood in the Bangladesh context, as is its impact on the fetus or neonate. In 2018, neonatal mortality in Bangladesh accounted for two-thirds of child mortality (3% of live births). Perinatal mortality, or death occurring from 28 weeks' gestation through the seventh day of life, occurred in 4.8% of pregnancies in Bangladesh, of which 52% were stillbirths [8, 9].

Women are encouraged to seek early and regular antenatal care from a medically qualified provider to ensure a healthy pregnancy and safe birth [10]. In Bangladesh, between 2014 and 2018, the percentage of women receiving any antenatal care from a medically qualified provider increased from 64% to 82% [8, 11]. In 2018, 42% of women in rural areas received four or more antenatal care contacts during pregnancy, although this increase in care seeking has not corresponded with a reduction in perinatal mortality [8, 12]. The objective of this study was to investigate the association between maternal hypertension and the risk of perinatal mortality in rural Bangladesh, and whether that association was mediated by antenatal care.

## Materials and methods

### Study setting and study design

This nested, matched case-control study was a sub-study of the Child Health and Mortality Prevention Surveillance (CHAMPS) project site in the Baliakandi sub-district of Bangladesh. CHAMPS Bangladesh began active population-based demographic surveillance in September 2017 in Baliakandi on a population of approximately 220,000. The details of CHAMPS methods have been published elsewhere [13, 14]. Ethical approval for the study was provided by the Ethical Review Committee of the International Centre for Diarrhoeal Disease Research, Bangladesh.

**Cross sectional survey.** From April to August 2019, we conducted a single survey among married women of reproductive age in the CHAMPS Baliakandi demographic surveillance system. The survey tool was based on the Demographic and Health Survey (DHS) Program and the WHO STEPwise approach to surveillance surveys [15, 16]. The questions were translated into Bengali and validated through prior national surveys [8, 9, 11, 17]. All married women of reproductive age living in households with a child (living or dead) under five years of age, or a woman currently pregnant or pregnant within the previous 12 months, were eligible to participate. One week prior to the start of data collection in each block of the demographic surveillance system, a listing of households meeting the eligibility criteria was generated using CHAMPS data. Data collectors conducted face-to-face interviews after taking written informed consent from the woman. If the woman was under 18 years of age, informed assent was taken and witnessed by a guardian from the household. Data collectors conducted up to nine follow-up visits to complete the data collection, with women self-reporting previous screenings, diagnoses of diabetes and hypertension and timing of any diagnoses, the numbers of antenatal care contacts they had during current and recent pregnancies, and the location, type of health care provider, and services rendered. No medical records were available to confirm the self-reported information.

**Demographic surveillance databases from parent cohort study.**   We used the CHAMPS demographic surveillance database dated February 26, 2020 to identify all singleton pregnancy outcomes among survey respondents to include those outcomes which took place up to one year before or 14 days after the date of the survey. We only considered singleton pregnancies for this analysis, as multiparity is an independent risk factor for perinatal death [12, 18]. We extracted demographic, socio-economic and pregnancy history information for each woman with an eligible pregnancy from the CHAMPS demographic surveillance database by linking unique identification numbers.

**Outcome: Perinatal death.**   Perinatal death was defined as a pregnancy ending in the death of the offspring between the completion of 28 weeks gestation and seven days following delivery [19]. Deaths occurring after 28 weeks of pregnancy, but before or during birth were classified as a stillbirth; those occurring between 0 and 7 completed days after birth were classified as early neonatal deaths [19].

## Exposure: Hypertension

- Any hypertension: Any self-reported diagnosis of hypertension, regardless of timing.

- Presumed pre-gestational (chronic) hypertension: Self-reported diagnosis of hypertension at any time prior to the index pregnancy.

- Presumed gestational hypertension: Self-reported diagnosis of hypertension during or after the index pregnancy, in the absence of any self-reported previous diagnosis of hypertension. We assumed that hypertension diagnosed post-partum was likely due to unresolved hypertension that was present during pregnancy.

## Data analysis

**Selection of cases and controls.**   Cases of pregnancies ending in a perinatal death were identified by comparing the date of pregnancy outcome and date of the woman's last menstrual period preceding the pregnancy. Eligibility to serve as a matched control was defined as a pregnancy ending in a live birth in which the neonate survived beyond seven days, such a pregnancy being identified by the absence of a reported death within the first seven days of life in the CHAMPS demographic surveillance system. Among cases and eligible controls, we excluded pregnancies in women who reported no previous screening for hypertension and pregnancies for which we had no antenatal care information. For women who had more than one eligible pregnancy, we selected the pregnancy with an outcome closest to the survey date.

**Matching.**   Based on a literature review of confounders for the relationship between hypertensive disorders of pregnancy and perinatal mortality we matched on the following characteristics: age group (<20, 20–24, 25–29, 30–34, 35–39, 40+), wealth quintile, and educational attainment (none, primary, secondary, post-secondary) [8, 20–25]. The wealth quintile was constructed using the DHS wealth index score [26, 27]. We randomly matched, without replacement, each case with up to five controls. For cases with fewer than five matched controls, we conducted the analysis with the controls as matched.

Using summary statistics and chi-squared tests, we examined the differences in demographic, hypertension, and pregnancy characteristics between cases and controls. We then used conditional logistic regression in the matched case-control groups to estimate probability ratios for perinatal death by any hypertension diagnosis, nature of hypertension, and number of antenatal care contacts with a qualified provider (0, <4, 4+) for the index pregnancy. Looking at median and interquartile ranges for antenatal care timing and frequency, we explored

the relationship between antenatal care-seeking, the type of hypertension, and perinatal death. P<0.05 was considered statistically significant for all analyses.

## Results

A total of 4,550 pregnancies met the eligibility criteria for inclusion in the study, including 201 perinatal deaths (Fig 1). The overall perinatal mortality rate was 44 per 1,000 pregnancies (4.4%), which was consistent with national surveys [8, 11]. Among women with eligible pregnancies, 97.5% reported screening for hypertension at least once in their lifetimes. Of those

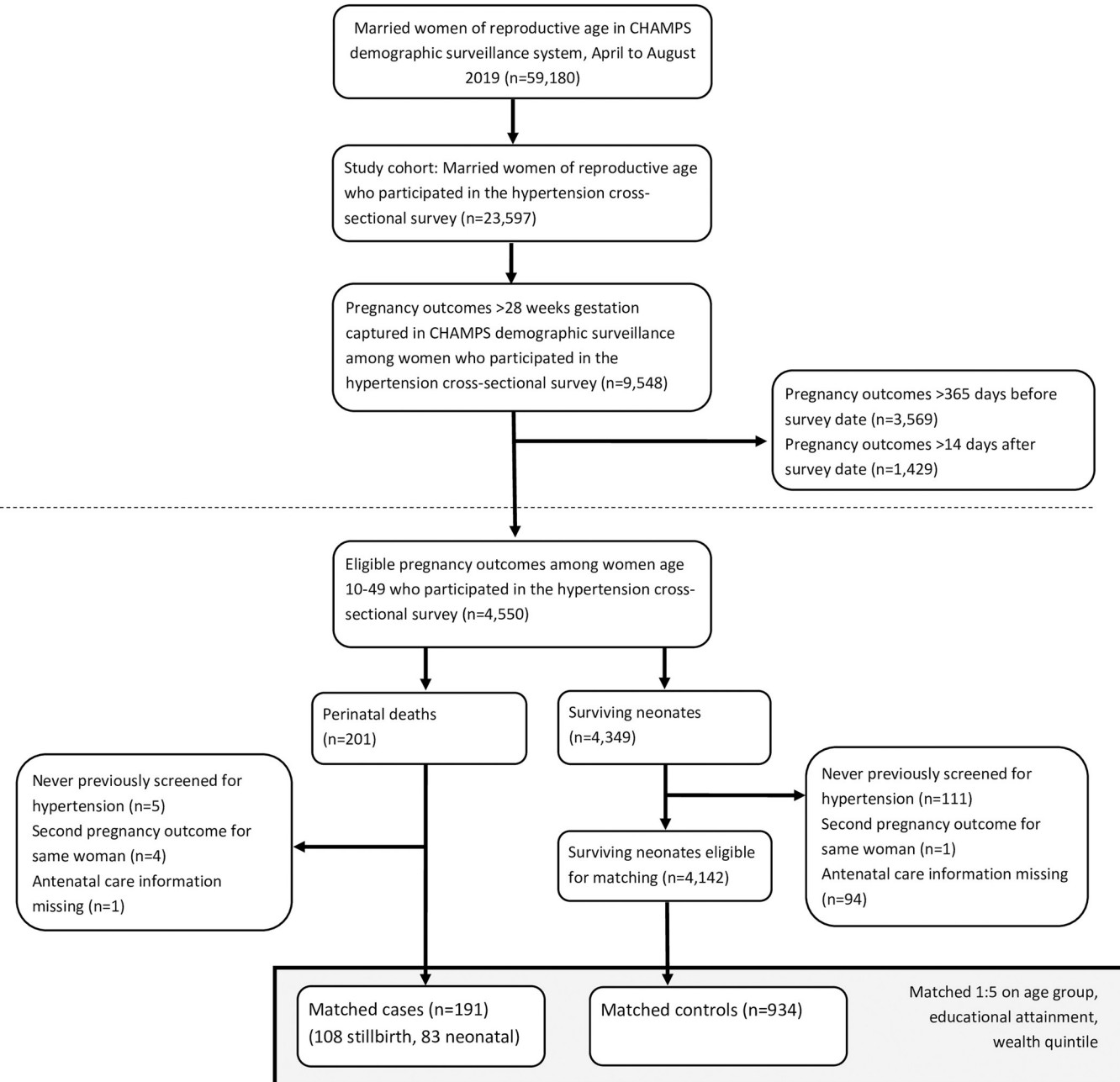

**Fig 1. Sample selection of women, nested case-control study, hypertension and perinatal mortality in Baliakandi, Bangladesh, 2019.**

**Table 1. Comparisons of socio-economic characteristics, health history, and care seeking in pregnancy between cases and matched controls, Baliakandi, Bangladesh, 2019.**

| Characteristic | Cases of perinatal death (n = 191) | Matched Controls (n = 934) | P value |
|---|---|---|---|
| Mean maternal age in years at the time of birth | 23.6 | 23.8 | 0.75 |
| **Maternal Age in Years** | | | 1.00 |
| <20 | 55 (28.8) | 271 (29.0) | |
| 20–24 | 64 (33.5) | 318 (34.0) | |
| 25–29 | 32 (16.7) | 154 (16.5) | |
| 30–34 | 28 (14.7) | 138 (14.8) | |
| 35–39 | 12 (6.3) | 53 (5.7) | |
| **Education Completed** | | | 1.00 |
| None | 13 (6.8) | 53 (5.7) | |
| Primary | 47 (24.6) | 226 (24.2) | |
| Secondary | 91 (47.6) | 455 (48.7) | |
| Post-Secondary | 40 (20.9) | 200 (21.4) | |
| **Household Wealth Quintile** | | | 1.00 |
| Lowest | 42 (22.0) | 208 (22.3) | |
| Second | 38 (20.0) | 186(19.9) | |
| Middle | 47 (24.6) | 222 (23.8) | |
| Fourth | 40 (20.9) | 198 (21.2) | |
| Highest | 24 (12.6) | 120 (12.8) | |
| **Timing of Perinatal Death** | | | |
| Stillbirth | 108 (56.5) | NA | |
| Early Neonatal Death | 83 (43.5) | NA | |
| **Any Hypertension Diagnosis** | 27 (14.1) | 72 (7.7) | <0.01* |
| **Nature of Hypertension** | | | <0.01* |
| None | 164 (85.9) | 862 (92.3) | |
| Presumed pre-gestational hypertension | 10 (5.2) | 19 (2.0) | |
| Presumed gestational hypertension | 17 (8.9) | 53(5.7) | |
| **Antenatal Care Contacts with a Qualified Provider** | | | 0.80 |
| None | 36 (18.5) | 192 (20.1) | |
| <4 | 115 (60.2) | 562 (60.2) | |
| 4 or more | 40 (20.9) | 180 (19.3) | |

\* denotes significance at the p<0.05 level

screened, 415 (9.4%) reported a diagnosis of hypertension. A total of 216 (4.7%) pregnancies met exclusion criteria, leaving 4,333 unique women with pregnancies eligible for matching, including 191 cases of perinatal death (Fig 1). After matching, 10 cases had greater than one but fewer than five matched controls. The final analysis included 191 cases and 934 matched controls (Fig 1). There were no cases of perinatal death among 19 pregnancies in women over 40 years of age. This age group was dropped from the analysis.

Among women who experienced a perinatal death, 79% (151/191) were under 30 years of age, and over two-thirds (69%, 131/191) had completed secondary education or higher (Table 1). There were fewer cases of perinatal death among women in the highest wealth quintile (12.6%) compared with lower wealth quintiles. Slightly more than half (56%) of perinatal deaths were classified as stillbirths (Table 1). Women who experienced a perinatal death were more likely to report being diagnosed with hypertension compared with controls (14.1% vs.

7.7%, p<0.01), and to have pre-gestational high blood pressure (5.2% vs. 2.0%, p = 0.01). The proportion of women who had no contact with a qualified health care provider during pregnancy was similar between cases (18.5%) and controls (20.1%) (Table 1).

## Hypertension and perinatal death

Women who reported any hypertension diagnosis were more likely to experience perinatal death, and risk of perinatal death differed by timing of hypertension diagnosis. Women with pre-gestational hypertension experienced a 2.9-fold greater risk of perinatal death (OR 2.90, 95% CI 1.29, 6.57) than women who reported no diagnosis. Women with presumed gestational hypertension experienced a 68% increased probability of perinatal death (OR 1.68, 95% CI 0.98, 2.98) than women who reported no diagnosis, but this was not statistically significant.

**Antenatal care service contacts.** We found no association between the number of antenatal care contacts with a qualified provider and perinatal death (Table 2). Among women who experienced a perinatal death, 81% sought antenatal care at least once from a medically qualified provider. We evaluated the frequency and timing of antenatal care among women with any hypertension diagnosis by perinatal outcome (Fig 2). Across all categories of perinatal outcome and hypertension status, the total number of contacts with qualified health care providers in each pregnancy was below both the national standard of four and the WHO recommendation of eight antenatal care visits [10]. Women with presumed gestational hypertension had a higher median number of contacts with medically qualified providers (median 3, IQR 1,4) than women with presumed pre-gestational hypertension (median 2, IQR 1,3) and women never diagnosed with hypertension (median 2, IQR 1,3), and this difference was statistically significant (p = 0.02). The median number of contacts with a qualified health care provider did not differ significantly by perinatal outcome within hypertension groups (p = 0.66).

Among women who sought any care during pregnancy, 33% (339/1,023) obtained their first antenatal service from a non-qualified provider. Most of these service contacts (90%, 306/339) took place at home in median month four (IQR 2,5) or at a satellite clinic close to home at median month three (IQR 3,5). Among women who sought care with a medically qualified provider, the median first antenatal care contact was in the second trimester of pregnancy in all groups (Fig 3). We found no difference in median timing of first contact with a medically qualified provider by pregnancy outcome among women with no history of hypertension (p = 0.42) (Fig 3). Women with presumed pre-gestational hypertension who experienced a

**Table 2. Conditional logistic regression estimating univariate associations of hypertension status and antenatal care with perinatal death, matched on age group, educational attainment, and wealth quintile, Baliakandi, Bangladesh, 2019.**

| Characteristic | Matched Odds Ratio (95% CI) |
|---|---|
| **Any Hypertension Diagnosis** | 1.97 (1.23,3.19)* |
| Nature of Hypertension | |
| None | Ref |
| Presumed pre-gestational hypertension | 2.90 (1.29,6.57)* |
| Presumed gestational hypertension | 1.68 (0.95,2.98) |
| **Antenatal Care Contacts with a Qualified Provider** | |
| None | Ref |
| <4 | 1.11 (0.73,1.67) |
| 4 or more | 1.22 (0.73,2.02) |

* denotes significance at the p<0.05 level

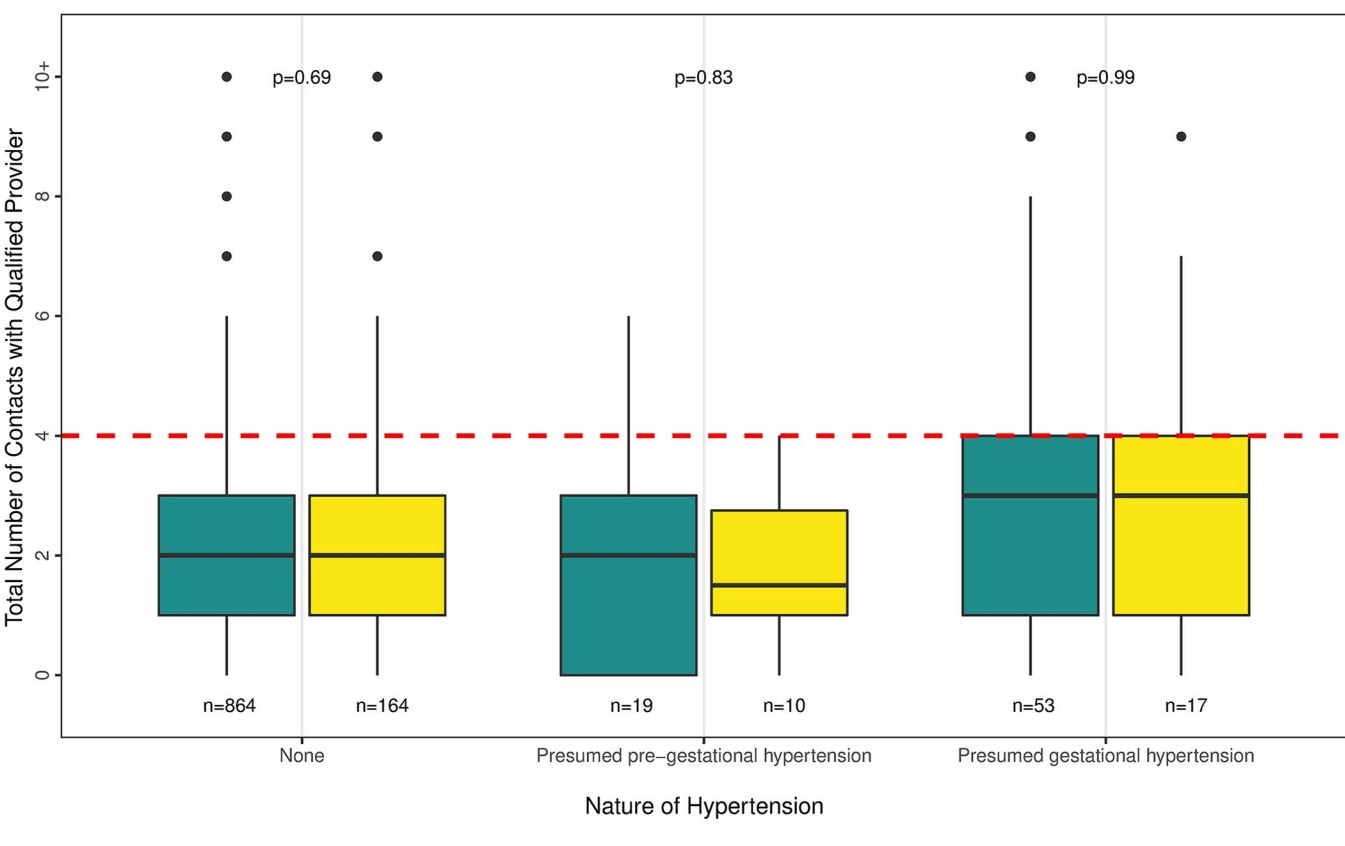

**Fig 2. Number of antenatal service contacts by nature of hypertension and pregnancy outcome, Baliakandi, Bangladesh, 2019 (n = 1,125).**

perinatal death had their median first contact with a qualified provider later than any other group, in the sixth month of gestation (IQR 6, 8) (Fig 3), and 89% of these women sought this care in a health care facility (clinic or hospital). There was high variation in timing of first antenatal service contact within almost all groups, the exception being the group of women with presumed pre-gestational hypertension who experienced a perinatal death, 78% (7 out of 9) of which had their first antenatal service contact with a qualified care provider in the sixth or seventh month of pregnancy.

## Discussion

This retrospective case-control study of 1,125 women found that any pre-pregnancy history of hypertension was a significant risk factor for perinatal death in Baliakandi sub-district, Bangladesh. Antenatal care was suboptimal in timing and frequency among all groups of women regardless of hypertension status or pregnancy outcome, and the frequency of antenatal care-seeking was not associated with fetal or newborn survival.

Chronic hypertension in younger women has not been well studied in Bangladesh. The first national hypertension prevalence survey of women 18–34 years of age was published in 2020 [8]. Pre-eclampsia is a well-recognized complication of pregnancy in Bangladesh, and previous studies have found that pre-eclampsia contributes to 24% of maternal mortality, 7% of still-births, and 2.9% of neonatal deaths [12, 28, 29]. Khanam et. al found that probable pregnancy-induced hypertension was a significant risk factor for stillbirth (IRR = 1.8, 95% CI 1.3–2.5) in

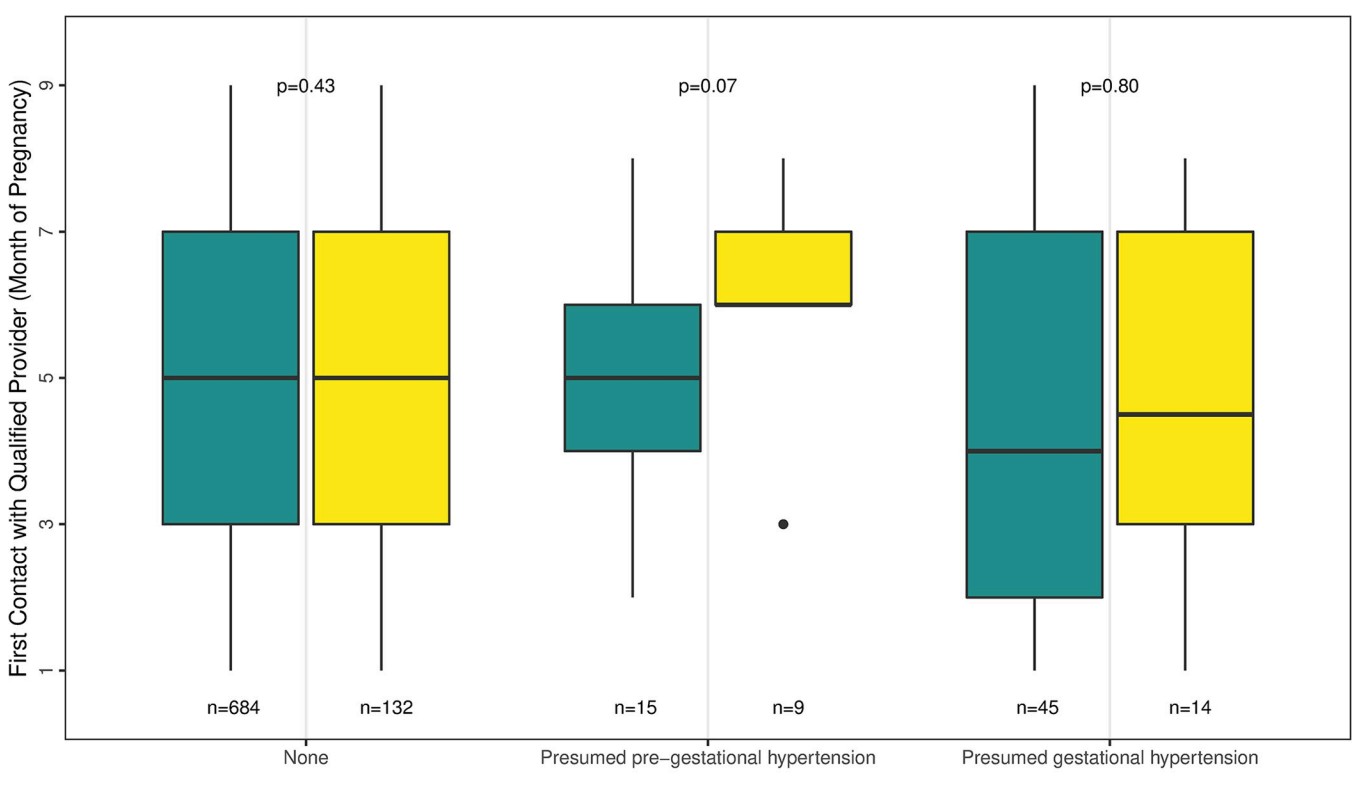

**Fig 3. Month of first contact with a medically qualified provider during pregnancy, by nature of hypertension and pregnancy outcome, Baliakandi, Bangladesh, 2019 (n = 897).**

Sylhet, Bangladesh [30]. This study suggests that underlying chronic high blood pressure may be contributing more significantly to perinatal death than has been previously recognized in Bangladesh. Future studies looking at the etiology of perinatal death should attempt to better differentiate between underlying chronic disease and pregnancy-induced disease in order to inform the development of effective public health and medical interventions for preventing adverse pregnancy outcomes.

In all rural areas of Bangladesh, 21% of women seek no antenatal care in pregnancy, and only 20% are seen four or more times by a qualified health care provider [8]. Our study found similar antenatal care-seeking patterns among women in Baliakandi [8]. Baliakandi women in our study who experienced a perinatal death had the same median number of antenatal care visits (three) from a qualified antenatal care provider as rural women elsewhere in Bangladesh, with the median timing of first contact with a qualified health care provider taking place in the fourth month of pregnancy [29]. Our study suggests that this timing and frequency of antenatal care is insufficient to prevent adverse perinatal outcomes. Additional research is needed to understand why this contact with the health care system did not have a protective effect. Possible explanations could be related to poor quality of care by the health care provider, late onset of care-seeking, or weak referral systems once complications were identified [31–34].

Women with a history of hypertension predating pregnancy and who experienced a perinatal death sought care late in their pregnancies, possibly spurred by the onset of hypertension-

induced complications. Further qualitative research is needed to understand why women with pregnancies complicated by hypertension, including women with a known history of disease, did not have more frequent contact, as per national guidelines, with the health care system in order to effectively manage their high-risk pregnancy [35]. Focused efforts to identify women with a history of disease and encourage them to seek early and regular antenatal care from a hospital-level facility equipped to treat high-risk pregnancies could be an effective intervention for improving outcomes in this high-risk group, but only if those facilities are equipped with the personnel, equipment, supplies, and essential medicines to effectively manage hypertension in pregnancy [2, 6, 34].

A limitation of this study is the potential for misclassification due to the absence of direct measurement or medical records to confirm self-reported information on hypertension diagnosis and antenatal care-seeking. We posit that a woman's ability to correctly recall a pre-pregnancy diagnosis of hypertension is independent of perinatal outcome, in which case misclassification would result in our findings being an underestimate of the true magnitude of the association between hypertension and perinatal death. It is possible, in the effort to understand the causes of adverse events, that women who experienced a perinatal death were more likely to be correctly identified as hypertensive than those with a healthy outcome, leading to differential misclassification by outcome and an overestimate of the association between hypertension diagnosis, either during or after pregnancy, and perinatal death. The marginal association between presumed gestational hypertension and perinatal death (OR 1.68, 95% CI 0.98, 2.98) suggests a clear association that would have been statistically significant with a larger sample size. These data provide a baseline for future longitudinal research to both investigate the temporality of hypertension and further elucidate how hypertension affects pregnancy outcomes throughout a woman's childbearing years in rural, resource-constrained settings.

The Government of Bangladesh is committed to reducing the neonatal mortality rate to 12 deaths per 1,000 live births by 2030 [36]. In order to achieve both this goal and that of similar reductions in stillbirths, greater emphasis should be placed on identification of women with a history of hypertension and improving their access to and utilization of skilled antenatal and delivery care. Consistent with national guidelines, antenatal care for women with pre-gestational hypertension should be early, frequent, and take place in hospital settings with the trained personnel, equipment, supplies, and medicines necessary for effective management of complicated pregnancies [35]. Raising community awareness of the dangers of hypertension in pregnancy, especially when there is a diagnosis of hypertension prior to pregnancy, should be included in any programs introduced to address low care-seeking among these high-risk women [35].

## Author Contributions

**Conceptualization:** Allyson P. Bear, Wendy L. Bennett, Joanne Katz.

**Data curation:** Allyson P. Bear, Kyu Han Lee, Atique Iqbal Chowdhury.

**Formal analysis:** Allyson P. Bear, Kyu Han Lee, Emily S. Gurley.

**Funding acquisition:** Sanwarul Bari, Shams El Arifeen, Emily S. Gurley.

**Investigation:** Atique Iqbal Chowdhury, Sanwarul Bari, Shams El Arifeen, Emily S. Gurley.

**Methodology:** Allyson P. Bear, Kyu Han Lee, Sanwarul Bari, Emily S. Gurley.

**Project administration:** Atique Iqbal Chowdhury, Shams El Arifeen.

**Resources:** Sanwarul Bari, Shams El Arifeen.

**Software:** Kyu Han Lee, Sanwarul Bari.

**Supervision:** Sanwarul Bari, Shams El Arifeen, Emily S. Gurley.

**Validation:** Wendy L. Bennett, Joanne Katz.

**Visualization:** Kyu Han Lee.

**Writing – original draft:** Allyson P. Bear.

**Writing – review & editing:** Allyson P. Bear, Wendy L. Bennett, Joanne Katz, Atique Iqbal Chowdhury, Sanwarul Bari, Shams El Arifeen, Emily S. Gurley.

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
