## [Decision Letter · Decision Letter 0]

13 Oct 2023

PONE-D-23-17562Associations of hypertension and antenatal care-seeking with perinatal mortality: A nested case-control study in rural BangladeshPLOS ONE

Dear Dr. Bear,

Thank you for submitting your manuscript to PLOS ONE. After careful consideration, we feel that it has merit but does not fully meet PLOS ONE’s publication criteria as it currently stands. Therefore, we invite you to submit a revised version of the manuscript that addresses the points raised during the review process.

We look forward to receiving your revised manuscript.

Kind regards,

Abera Mersha, MSc.

Academic Editor

PLOS ONE

Journal Requirements:

Reviewers' comments:

Reviewer's Responses to Questions

**Comments to the Author**

1. Is the manuscript technically sound, and do the data support the conclusions?

Reviewer #1: Yes

Reviewer #2: Yes

2. Has the statistical analysis been performed appropriately and rigorously? 

Reviewer #1: Yes

Reviewer #2: Yes

3. Have the authors made all data underlying the findings in their manuscript fully available?

Reviewer #1: No

Reviewer #2: Yes

4. Is the manuscript presented in an intelligible fashion and written in standard English?

Reviewer #1: Yes

Reviewer #2: Yes

5. Review Comments to the Author

Reviewer #1: General Comment:

This study presents a nested matched case-control analysis, utilizing a cross-sectional survey and a demographic database in Bangladesh, to examine the relationship between maternal hypertension and perinatal mortality in rural Bangladesh. The authors propose that a history of hypertension before pregnancy significantly increases the risk of perinatal mortality, acknowledging, however, the lack of direct measurements or medical records to verify self-reported hypertension diagnoses and antenatal care visits. The paper clearly outlines its research procedures and limitations, thereby providing useful insights for readers. Additionally, including interviews with women from rural Bangladesh adds significant value to the study.

Additional Comments:

1. Methods, Cross-sectional survey (L.50-65)

The authors gathered data through interviews instead of questionnaires, despite translating and validating the questions. Given the reference to qualitative research in this paper, it is presumed that the authors considered using qualitative data collection methods during the research design phase. These might include using open-ended questions in interviews or having researchers make field notes during follow-up visits. Such data could offer valuable insights into the participants' perceptions and behaviors versus the reality in Bangladesh and form the foundation for the authors' argument.

2. Discussion (L.204-227)

The authors address the insufficiency of current maternal care in preventing adverse perinatal outcomes and scrutinize why maternal antenatal care behaviors have not effectively functioned as preventive measures (L.199-203). They also hypothesize about women who experienced perinatal deaths and had pre-existing hypertension before pregnancy (L.204-206). Would the interviewers' experiences not be relevant to these causal explanations and assumptions? The reviewer understands that the paper is presented within the framework of an epidemiological study, that is, a quantitative study. However, integrating qualitative research into a quantitative framework could further enhance this study.

Please note that these comments do not constitute a request for manuscript revisions. Thank you for the opportunity to review this quality research paper.

Reviewer #2: Please clarify this DOI https://doi.org/10.1101/2023.06.13.23291331.

What influence did culture and accessibility have on your study variables (health-seeking and perinatal mortality, respectively)?

What are your recommendations for further studies?

6. PLOS authors have the option to publish the peer review history of their article (what does this mean?). If published, this will include your full peer review and any attached files.

Reviewer #1: No

Reviewer #2: No

---

## [Author Response · Author response to Decision Letter 0]

29 Oct 2023

Thank you to the reviewers for the time that you have taken to review the manuscript and insightful comments made. Your careful consideration is much appreciated. Specific responses are addressed in the "response to reviewers" document included in this resubmission.

---

## [Decision Letter · Decision Letter 1]

9 Jul 2024

Associations of hypertension and antenatal care-seeking with perinatal mortality: A nested case-control study in rural Bangladesh

PONE-D-23-17562R1

Dear Dr. Bear,

We’re pleased to inform you that your manuscript has been judged scientifically suitable for publication and will be formally accepted for publication once it meets all outstanding technical requirements.

Kind regards,

Sayedur Rahman, MBBS, MMSc

Academic Editor

PLOS ONE

Additional Editor Comments (optional):

The authors have adequately addressed the comments raised by both reviewers. In my opinion, the revised manuscript is technically and scientifically suitable for publication.

Reviewers' comments:

Reviewer's Responses to Questions

**Comments to the Author**

1. If the authors have adequately addressed your comments raised in a previous round of review and you feel that this manuscript is now acceptable for publication, you may indicate that here to bypass the “Comments to the Author” section, enter your conflict of interest statement in the “Confidential to Editor” section, and submit your "Accept" recommendation.

Reviewer #1: (No Response)

2. Is the manuscript technically sound, and do the data support the conclusions?

Reviewer #1: (No Response)

3. Has the statistical analysis been performed appropriately and rigorously? 

Reviewer #1: (No Response)

4. Have the authors made all data underlying the findings in their manuscript fully available?

Reviewer #1: (No Response)

5. Is the manuscript presented in an intelligible fashion and written in standard English?

Reviewer #1: (No Response)

6. Review Comments to the Author

Reviewer #1: (No Response)

7. PLOS authors have the option to publish the peer review history of their article (what does this mean?). If published, this will include your full peer review and any attached files.

Reviewer #1: No

---

## [Editor Report · Acceptance letter]

11 Jul 2024

PONE-D-23-17562R1 

PLOS ONE

Dear Dr. Bear, 

I'm pleased to inform you that your manuscript has been deemed suitable for publication in PLOS ONE. Congratulations! Your manuscript is now being handed over to our production team.

Kind regards, 

on behalf of

Dr. Sayedur Rahman 

Academic Editor

PLOS ONE